# Status and Analysis of Artificial Breeding and Management of Aquatic Turtles in China

**DOI:** 10.3390/biology11091368

**Published:** 2022-09-17

**Authors:** Xiaoyou Hong, Xiaoyan Zhang, Xiaoli Liu, Yakun Wang, Lingyun Yu, Wei Li, Fangcan Chen, Xinping Zhu

**Affiliations:** 1Key Laboratory of Tropical & Subtropical Fishery Resource Application & Cultivation of Ministry of Agriculture, Pearl River Fisheries Research Institute, Chinese Academy of Fishery Sciences, Guangzhou 510380, China; 2China Wildlife Conservation Association Aquatic Wildlife Protection Branch, Beijing 100000, China; 3Guangzhou Qianjiang Water Ecology Technology Co., Ltd., Guangzhou 510380, China

**Keywords:** artificial breeding, management, census, aquatic turtles, China

## Abstract

**Simple Summary:**

Turtles have been on Earth for 300 million years, but their survival has been challenged as human activity has expanded. The results of a census and statistical analysis of artificially domesticated aquatic turtles in 15 provinces of China showed that 29 species were aquatic turtles native to China and a large number of exotic aquatic turtles are domesticated in China. This paper presents the current situation and problems of aquaculture and conservation of aquatic turtles in major provinces of China, and puts forward some suggestions for the conservation and management of aquatic turtles.

**Abstract:**

China is a major country in turtle cultivation and has a long history of artificial breeding of turtles. In this study, a census and statistical analysis of artificially domesticated aquatic turtles in 15 provinces of China were conducted. The results showed that 29 species were aquatic turtles native to China, accounting for approximately 9% of the world’s total, and a large number of exotic aquatic turtles are also domesticated in China. The present situation of artificial breeding and protection of aquatic turtles in major provinces of China is shown, and existing problems are also put forward, with suggestions for the protection and management of aquatic turtles.

## 1. Introduction

Turtles have been on earth for 300 million years, but with the expansion of human activities, their survival is facing challenges. There are 36 species in six families in China, including five sea turtles and three tortoises, out of approximately 330 species in 14 families worldwide [1,2,3]. Because of the turtle culture in China, artificial breeding of turtles was initiated, and the history of turtle culture can be traced back to prehistoric times. In the Shang Dynasty, the turtle shell was already used as a divination tool. Now China’s breeding turtles are mainly used for food, medicine, and ornamental and wild proliferation. The Law of the People’s Republic of China on the Protection of Wildlife was implemented in 1989. The current effective version is the fourth revision, passed in 2018, and it aims to protect wild animals, save precious and endangered wild animals, maintain biodiversity and ecological balance, and promote ecological civilization-building laws.

The Regulations on the Implementation of the Protection of Aquatic Wildlife of the People’s Republic of China is a law that was issued in 1993. After two revisions in 2011 and 2013, it clearly stipulates that people who domesticate and breed aquatic wildlife under state protection should hold domestication and breeding licenses issued by relevant fishery administrative departments. The Aquatic Wildlife Licensing Measures of the People’s Republic of China is a law that was issued in 1996 and that has been amended several times, in 2004, 2010, and 2013, to regulate the issuance and use of the licensing certificates for the utilization of aquatic wildlife; this license is called the Aquatic Wildlife Domestication and Breeding License and Aquatic Wildlife Operation and Utilization License. China is a major country in the domestication and reproduction of turtles and tortoises. Since the implementation of the domestication permit policy in 1996, it is still unknown what the current management and protection status is. Based on survey data, this paper gives a general overview of the protection and management of aquatic turtles in China and proposes solutions to existing problems.

## 2. Materials and Methods

From 2019 to 2020, a census was sent to the fishery and fishery administration departments of 15 provinces and municipalities, including Anhui Province, Hubei Province, Guangdong Province, Shanghai City, et al. (Figure 1), to count the licenses for the use of aquatic wild turtles (units might have a breeding license, operating license, or both) and to count factors related to operations and utilization (the number of operating and utilization institutions in each province and municipalities, species types, total number of animals, and contact details).

The census forms collected in 15 provinces and municipalities were sorted, and questionable information was checked by telephone consultation to make the data as accurate as possible.

Turtles are classified according to the book *Turtles of the World* [1].

Using the fishery yearbook [4], the output of edible turtles (Chinese softshell turtle, *Pelodiscus sinensis*) in China from 2004 to 2021 was calculated.

## 3. Results

### 3.1. Turtle Species

According to government statistics, the turtle species that are certified for commercial rearing in China include all 12 families except the Pelomedusidae family (because the Testudinidae family does not belong to the fishery management department, no research has been done on this family), for a total of 88 species. Among the 88 species of turtles, 29 are native species in China, and 59 are exotic species. See Table 1 for details; see Table A1 for species diversity.

### 3.2. Certificate Status

A total of 28,862 artificial domestication, breeding, and business utilization units in 15 provinces and municipalities were surveyed. Among them, 17,007 units independently held artificial breeding licenses, and 489 units independently held operation and utilization licenses. There were 11,349 units that held two certificates; see Table 2.

### 3.3. Artificial Domestication and Breeding

The number of cultured turtles in the surveyed provinces is shown in Table 3. In addition to the three species of the Testudinidae family, 29 native turtle species were counted, and both the protected turtles and the unprotected Chinese softshell turtle have been artificially domesticated. Inferring from the amount of domesticated individuals, the two large turtles are the most endangered, and Swinhoe’s Softshell Turtle *Rafetus swinhoei* is the least endangered; the most endangered species had just one male in the units surveyed, and the Asian giant softshell turtle *Pelochelys cantorii* had just adult 13 individuals and nearly 1000 offspring (Figure 2).

There are a number of sea turtles held in parks for the public to view. The most endangered sea turtle is the leatherback turtle, *Dermochelys coriacea*, of which only two individuals were counted.

All species of the genus Cuora are protected species, and the number of *Cuora zhoui* is the lowest, 357, followed by *Cuora yunnanensis*, *Cuora mccordi*, *Cuora pani*, and *Cuora aurocapitata* with 1340, 4801, 5427, and 5476 individuals, respectively. Some domestication certificates show that there are as few as one domestic Cuora turtle in some units.

For the exotic turtles, there were 326,327 *Heosemys grandis*, 339,501 Podocnemis turtles, 247,714 *Cuora amboinensis*, 120,644 *Malaclemys terrapin*, and 86,954 *Geoclemys hamiltonii* (Figure 3).

### 3.4. Turtle Production

The total breeding volume of Chinese softshell turtles was 182,610 tons in 2004, 364,878 tons in 2021, and has been stable above 300,000 tons in the past 11 years. (Figure 4).

## 4. Discussion

### 4.1. Data Reliability Analysis

Most localities use reports combined with data to submit census results to describe the overall situation of the census in the province. The specific measures for the approval and management of the two licenses vary greatly from place to place, and the information recorded in the statistical tables varies, such as the approval number of the two licenses, the period of issuance, the validity period, the number of approved species, the number used in breeding, the number of individuals in the business, the purpose of the captive breeding, and the locations of sales. In some provinces, such as Shanghai and Yunnan, there are fewer than 10 units that are licensed for domestication, and there may be problems such as data leakage, so the real situation cannot be displayed. At the same time, there are also differences in the specific management of the annual review and renewal of the two certificates. For this reason, the same method cannot be used for statistical data, and individuals doing research need to consult and understand this situation. Even with a well-managed artificial breeding license, there is inevitably some level of uncertainty in the statistical data, so it can be used only as a management reference. However, the survey data can reflect the current situation of aquatic wildlife protection and management in China to a certain extent and provide a certain data basis for further improving the ledger [5], standardizing the approval of aquatic wildlife management and utilization, and combating illegal fishing and illegal management and utilization.

### 4.2. Conservation Management Status

Since the Wildlife Protection Law of the People’s Republic of China was issued and implemented in 1989, a series of laws and regulations on wildlife protection have been issued and revised [6]. These laws and regulations and the establishment of related management systems are aimed at improving the maintenance and management level of aquatic wildlife. With its vast territory, China has a large number of domesticated turtles and tortoises, and more than 28,000 households have obtained certificates, which has played an important role in the dynamic monitoring management, legal traceability and information sharing of aquatic wildlife and their products.

With the continuous deepening of turtle protection legislation, the power of law enforcement in this area has been strengthened. In particular, since the outbreak of COVID-19, national awareness of wildlife protection has increased, and law enforcement authorities have dealt with more cases of wild animal abuse and illegal trade, greatly enhancing the protection of turtles and other wild animals [7].

The national awareness of aquatic wildlife protection has been strengthened. The construction of a large number of science museums and national parks, as well as the development of aquatic animal protection publicity activities and aquatic animal proliferation and release activities, have greatly enhanced the national awareness of wildlife protection.

### 4.3. In Situ and Ex Situ Conservation

Artificial breeding plays an important and positive role in the conservation of endangered turtle populations. Aside from the failure of artificial assisted breeding with 2 *R. swinhoei* (one female and one male; the female turtle died during artificial induction in 2019) [8], other turtle species have been successfully artificially bred. More successful species, such as *Platuysternon megacephalum* [9] and *Mauremys mutica* [10], have played an important role in the growth of turtle populations.

Large-scale artificial breeding of turtles has been in China for nearly 40 years. Turtle breeding has played a huge role in people’s livelihood and economic development. The total annual supply of Chinese soft-shelled turtles is now stable at 300,000 tons per year, which has played an important role in the needs of people’s livelihood. According to the 2020 report of the Guangdong Turtle Breeding Industry Association, the number of employees involved in the artificial breeding of turtles was 265,600, and the direct economic output value exceeded 100 billion yuan in China’s Guangdong Province (unpublished).

Another important role of the artificial breeding of wild populations is to implement reintroduction to promote the recovery of wild populations. However, China is still lacking in the artificial release of endangered turtles and the restoration of wild populations. The main activities are the artificial release of *Chelonia mydas* [11], and some postrelease effect evaluations have been carried out. *M. mutica*, *P. sinensis*, and other species have also been artificially released in the wild many times. However, the monitoring and effect evaluation after the release are insufficient, and the effect of the release on the recovery of the wild population is unknown. In recent years, the protection work of China’s first-class national protected animal *P. cantorii* has had much influence on the protection of turtles across the country. There are 13 known wild *P. cantorii* individuals in China, and they were domesticated in six different areas [12]. Four of these individuals have been successfully bred (for seven consecutive years), and nearly 1000 offspring are currently alive. To better protect this national first-class protected animal, the state formulated and passed the “*Pelochelys cantorii* Conservation Action Plan (2019–2035)” and carried out the first breeding–release program for the species in 2020, with 20 artificially hatched individuals aged 4–5 years [13,14,15]. According to the growth data and the health status of the *P. cantorii* individuals that were recaptured in 2021 and 2022, the released individuals adapted to their wild environment; therefore, the wild adaptation and protection effect is remarkable, and artificial release in suitable historical habitats can be considered. China has further carried out habitat assessment, restoration, and transformation of the historical habitats of the species, carried out large-scale artificial release, and finally achieved the goal of wild population recovery [16].

### 4.4. Management

The usage of species names on the certificate is not standardized. The scientific name is often not used; instead, the species names on certificates have many irregularities: they may be local common names (such as Vietnamese stone turtles), abbreviations (such as for southern turtles), only order or genus names (such as for Turtles, *Cuora* turtles, sea turtles, and the South American side-necked turtle), or hybrid names (e.g., *Mauremys iversoni, Ocadia glyphistoma, Sacalia pseudocellata, Ocadia philippeni*) [17]. The species name convention on certificates needs to be standardized by national authorities, and at the same time, the training of local administrative staff should be strengthened.

There are also false numbers of captive individuals. Some rare turtles, such as box turtles, have a certain number of false reports from the statistical data and verification results. For example, one farm’s domestication certificate stated that the number of *Cuora yunnanensis* was 1000, but telephone verification revealed that there were actually only a few dozen individuals. The large number reported by the farmer is the index number for the later introduction and transaction of future *Cuora* turtles. Therefore, relevant government departments need to strengthen the on-site verification of the approval process to prevent false reports.

The skill level of turtle domestication technicians is uneven. The domestication certificate management agency lacks an effective review method for the technical level of the breeding personnel of the unit applying for the domestication certificate. There are some tasks for which farmers have no trained technical personnel; instead, these farmers use technical personnel from other units to obtain the qualifications for the certificate. The end result will be the death of some turtles belonging to rare species due to low skill levels and poor facilities.

During the investigation, it was also found that a relatively important problem was that there are many very small groups of endangered and rare species. For example, some *Cuora* turtles were represented by only one individual in some units, and in some breeding farms only mature females (and no suitable males). Due to policy and for other reasons, there is a lack of a communication mechanism between units that possess critically endangered turtles, and this prevents the effective integration of resources for increasing the population of endangered species.

To further improve the level of aquatic wildlife management, on 1 July 2021, the Ministry of Agriculture and Rural Affairs of the People’s Republic of China developed an “aquatic wildlife under special state protection information management system”, which started to be commissioned throughout the country, and gradually achieved the dynamic regulation of aquatic wildlife and its products, legal traceability, and information sharing. China will further strengthen the protection of aquatic wildlife. Species and habitats should be closely linked, but the management of aquatic species and nature reserves in China belongs to different management departments. The administrative structure of aquatic wildlife management is in the agricultural department; aquatic wildlife and nature reserves belong to the forestry department [18], and terrestrial turtles (Testudinidae) belong to the forestry department. There is a disconnection in management and a lack of coordination of powers and responsibilities, which results in inefficient management, hindrance of wildlife conservation, and ineffective enforcement of laws.

Most of the aquatic animal species that are designated as national key protected wild animals (this designation applies to only the wild populations), as found in the “Convention on International Trade in Endangered Species of Wild Animals” appendix, have been bred in captivity successfully. Some of these species are managed separately (for example, the wild original species are separately marked and managed in accordance with the Wild Animal Protection Law). Individuals that were born in captivity and that belong to more abundant species have substantial differences from wild populations, so they are managed in ways that are conducive to artificial breeding, circulation, and utilization, that reduce the difficulty of management, and that help promote resource protection. At the same time, China should also strengthen the training in and publicity of relevant laws and regulations, promote a positive understanding of the standardized management of aquatic wildlife utilization, and ultimately create a positive atmosphere for aquatic wildlife protection.

In terms of species, the survey data showed that exotic turtles accounted for 67.05%, which was similar to 68.29% in the survey of one of the largest turtle trading markets in South China [19]. Either escape or release of non-native turtles will cause irreversible negative impacts on China [20]. After alien turtles enter the wild, they have few or no natural enemies, and may occupy the ecological niche of native species, affect the production and reproduction of native species, and may also have a devastating impact on predators [21]. Alien species may also carry pathogens that native species are not resistant to, which will seriously affect the balance of native ecosystems [22]. At the same time, alien species may have gene exchange with native species, resulting in gene pollution [23]. Therefore, management measures against species invasion need to be more clear and proposed.

China is a major country in turtle breeding. Although there are certain problems in turtle breeding and protection management, the related work has been advancing in a positive direction. Among them, China’s artificial propagation technology for endangered tortoises, two-certificate management, aquatic wildlife information management system and label management can provide reference for tortoise species protection in other countries.

With the decision of the Standing Committee of the National People’s Congress in 2020 on comprehensively prohibiting illegal wildlife trade, eliminating the ecologically damaging habit of eating wild animals, and effectively protecting the lives, health, and safety of the people, the people’s awareness of wildlife protection has become increasingly strong. The management mechanism for the artificial breeding of turtles will become increasingly effective, and the protection and utilization prospects of aquatic turtles in China will be better.

## Figures and Tables

**Figure 1 biology-11-01368-f001:**
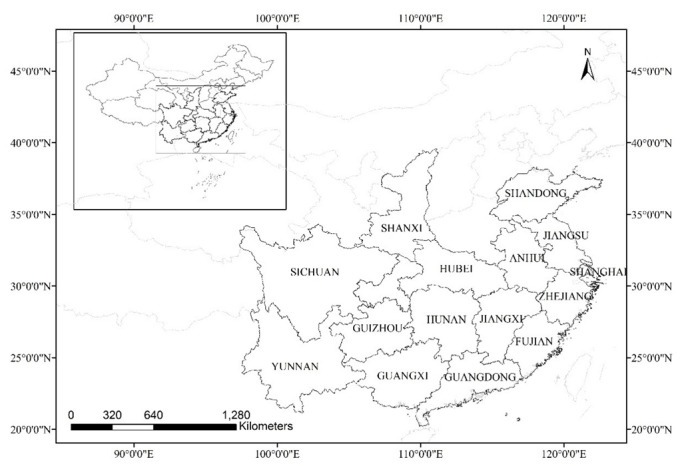
Provinces and municipalities surveyed in China.

**Figure 2 biology-11-01368-f002:**
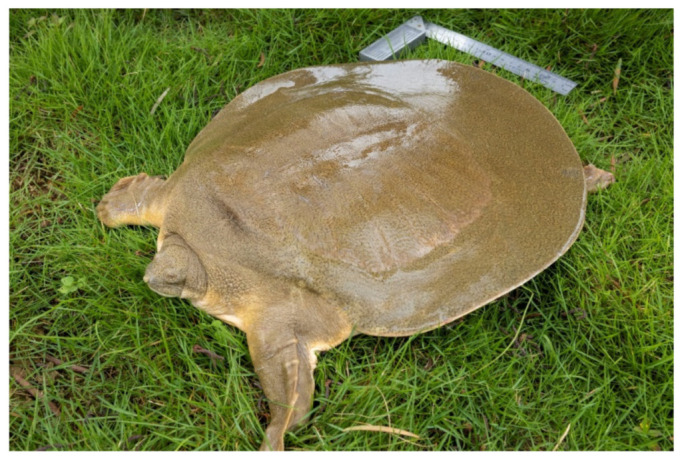
Artificially bred 5-year-old Asian giant softshell turtle.

**Figure 3 biology-11-01368-f003:**
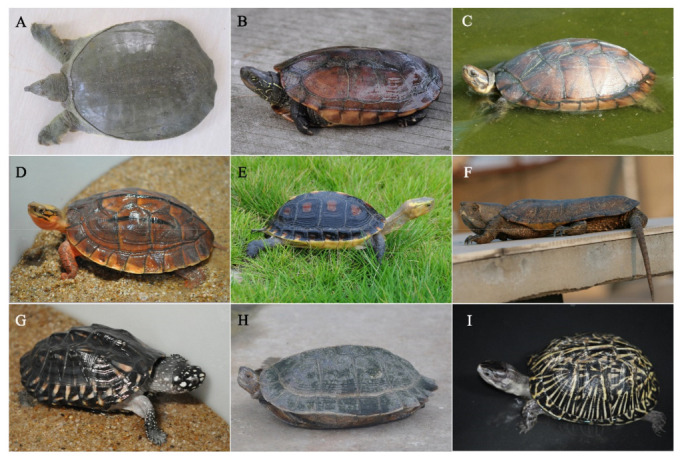
Some turtles domesticated in China. (**A**) Pelodiscus sinensis; (**B**) Chinemys reevesii; (**C**) Mauremys mutica; (**D**) Cuora trifasciata; (**E**) Cuora flavomarginata; (**F**) Platysternon megacephalum; (**G**) Geoclemys hamiltonii; (**H**) Heosemys grandis; (**I**) Terrapene Carolina.

**Figure 4 biology-11-01368-f004:**
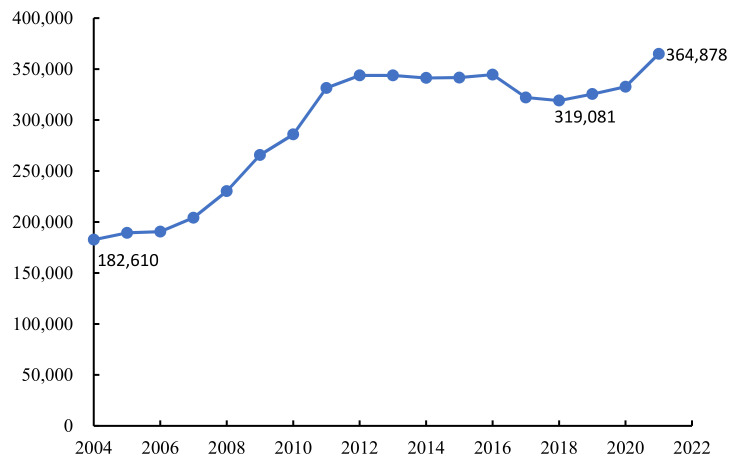
The output of Chinese soft-shelled turtles from 2004 to 2021 (unit: tons).

**Table 1 biology-11-01368-t001:** Number of genera/species of freshwater aquatic turtles in the surveyed provinces/municipalities.

Families	Guangdong	Anhui	Fujian	Guangxi	Guizhou	Hubei	Hunan	Jiangsu	Jiangxi	Shangdong	Sichuan	Yunnan	Zhejiang	Shangxi	Shanghai
Podocnemididae	2/2	1/1	3/3	1/1						1/1	1/1		2/5		
Chelidae	1/1		1/1	1/1							1/1				
Carettochelyidae	1/1		1/1							1/1					
Cheloniidae	4/4		2/2	3/3	1/1		2/2	4/4	2/2	3/3	3/3	2/2	3/3	3/3	
Chelydridae					1/1	1/1	1/1		1/1				2/2	1/1	
Dermatemydidae		1/1	1/1					1/1			1/1		1/1		
Dermochelydidae	1/1														
Emydidae	6/9	4/5	1/11	5/5	2/2	4/4	5/5	6/7	1/1	5/7	5/7		7/9		
Geoemydidae	15/38	9/24	16/37	8/25	2/2	3/9	5/13	10/25	4/12	14/31	10/27	5/8	12/30		2/3
Kinosternidae						2/4		2/3					1/1		
Platysternidae	1/1		1/1	1/1		1/1		1/1	1/1	1/1	1/1	1/1	1/1		
Trionychidae	7/7		1/2	1/2				1/1		2/2	2/2	2/2	6/6		1/1
Total	38/64	15/31	29/59	20/33	6/6	11/19	13/21	25/42	9/17	27/46	23/43	10/13	35/58	4/4	3/4

**Table 2 biology-11-01368-t002:** Number of certificate holders in the provinces/municipalities investigated.

Provinces/Municipalities	Total Number of Licensed Units	Number of Individual Captive Breeding License Holders	Number of Operating Permits Held Separately	Number of Certificate Holders with Two Certificates Held at the Same Time
Anhui	74	25	1	48
Fujian	58	30	1	27
Guangxi	4280	14	31	4218
Guizhou	6	2	1	3
Hubei	27	0	0	27
Hunan	13	1	2	10
Jiangsu	116	28	3	85
Jiangxi	27	0	0	27
Shandong	100	3	1	96
Shanxi	5	1	1	3
Shanghai	6	2	0	4
Sichuan	115	11	0	104
Yunnan	7	1	0	6
Zhujiang	207	68	13	126
Guangdong	23,820	16,821	435	6564
Total	28,861	17,007	489	11,348

**Table 3 biology-11-01368-t003:** The number of cultured turtles in various provinces/municipalities investigated.

Families	Guangdong	Anhui	Fujian	Guangxi	Guizhou	Hubei	Hunan	Jiangsu	Jiangxi	Shangdong	Sichuan	Yunnan	Zhejiang	Shangxi	Shanghai
Podocnemididae	4	50	2710	1005						240	710		21,801		
Chelidae	1126		120							10	50				
Carettochelyidae	9467		5790							25					
Cheloniidae	120		65	495	25		35	66	25	281	72	32	30	52	
Chelydridae					16	250	25		50				520	2	
Dermatemydidae		31	110					20			20		50		
Dermochelydidae	2														
Emydidae	160,960	684	12,210	10,453	30	855	92	740	6000	656	981		845		
Geoemydidae	34,515,876	282,713	89,283	5,095,966	4025	82,449	2760	102,880	138,889	8727	40,823	2668	83,577		140
Kinosternidae						485		280					35		
Platysternidae	10,270		1000	5471		170			5800	205	161	323	45		
Trionychidae	116,304		290	1,041,577				100		300	323,074	5160	35,540		2
Total	34,814,129	283,478	111,578	6,154,967	4096	84,209	2912	104,086	150,764	10,444	365,891	8183	142,443	54	142

## Data Availability

All datasets generated or analyzed during this study are included in the published article.

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
