# Peer review of "Status and Analysis of Artificial Breeding and Management of Aquatic Turtles in China"

_biology, 2022, doi:10.3390/biology11091368_

Round 1

Reviewer 1 Report

Dear authors,

Thanks for the opportunity in reviewing the manuscript “Status and analysis of artificial breeding and management of aquatic turtles in China” (Biology-1882854). This brief report presents a survey and relative discussion of artificially domesticated aquatic turtles in 15 provinces of China, the present situation of artificial breeding and protection of aquatic turtles in major provinces, and the existing problems. Overall, the manuscript is well written, and discussed an important practice and industry of utilization of wildlife resources in China. Please address my comments for minor revisions needed.

Minor:

Table 1: Please consider revising the title to ‘Number of genera/species of freshwater aquatic turtles in the surveyed provinces/municipalities’. And remove the repeated ‘genus/species’ in the second row.

Tables 2 and 3: Add commas in numbers. For example, replace “4280” with “4,280”.

Subheading 4.3: “In-situ and ex-situ’ for what? Consider specify in the subheading.

Please add some supporting citation for Sections 4.1 and 4.2.

Please add a paragraph of discussion regarding how the finding of this study can impact other relevant countries in the world.

Please add a paragraph in the Introduction with a brief background about why people breed these animals. In addition, please briefly describe how the law regulates such practices.  

Author Response

Point 1: Table 1: Please consider revising the title to ‘Number of genera/species of freshwater aquatic turtles in the surveyed provinces/municipalities’. And remove the repeated ‘genus/species’ in the second row.

Response 1: Accepted, thanks for the good suggestion.

Point 2: Tables 2 and 3: Add commas in numbers. For example, replace “4280” with “4,280”.

Response 2: Accepted, the numbers in the Tables 2 and 3 have been modified.

Point 3: Subheading 4.3: “In-situ and ex-situ’ for what? Consider specify in the subheading.

Response 3: What I meant to say was in situ or ex situ conservation for turtle, so “In situ and ex situ conservation” will be clear.

Point 4: Please add some supporting citation for Sections 4.1 and 4.2.

Response 4: Revised, three references were added.

Point 5: Please add a paragraph of discussion regarding how the finding of this study can impact other relevant countries in the world.

Response 5: One paragraph of discussion was added. “China is a big country of turtle breeding. Although there are certain problems in turtle breeding and protection management, the related work has been advancing in a positive direction. Among them, China's artificial propagation technology of endangered tortoises, two certificate management, aquatic wildlife information management system and label management can provide reference for tortoise species protection in other countries.’’ 

Point 6: Please add a paragraph in the Introduction with a brief background about why people breed these animals. In addition, please briefly describe how the law regulates such practices.  

Response 6:  “Because of the turtle culture in China, the artificial breeding of turtles was born, and the history of turtle culture can be traced back to the prehistoric times. In the Shang Dynasty, the turtle shell was already used as a divination tool. Now China's breeding turtles are mainly used for food, medicine, ornamental and wild proliferation” was added.

“the law regulates such practices“ was already stated in the second paragraph of the introduction.

Reviewer 2 Report

Dear Authors,

I read carefully your paper. I find it well written and easy to read and follow. In my opinion it is important contribution. I have following remarks and suggestions:

There are 88 species listed in your Appendix A table, whereas there are only 33 native species. In my opinion there should be included at least a short paragraph dealing with exotic species and possible danger if these species could establish stable wild populations (invasive species are serious problem in wildlife management). I suggest also to indicate native and exotic species - that would be useful for readers not familiar with Chinese fauna.

First two sentences of abstract do not fit together. Moreover, the first sentence is kind of eulogy of Chinese law. After reading this sentence one can make impression, that there is the most pro-environment country in the World, whereas there are numerous examples of ecologiacl disasters in China (but there are also spectacular successes). Plese revise.

Page 8. „Affairs of People's Republic China, 2020) . According to the growth data and the health” – there is space between parenthesis ‘)’ and a dot.

Author Response

Point 1: There are 88 species listed in your Appendix A table, whereas there are only 33 native species. In my opinion there should be included at least a short paragraph dealing with exotic species and possible danger if these species could establish stable wild populations (invasive species are serious problem in wildlife management). I suggest also to indicate native and exotic species - that would be useful for readers not familiar with Chinese fauna.

Response 1: After a detailed check, only 29 species of Chinese native turtles were investigated and counted, and the whole article was checked and revised. A statement about alien species and possible dangers has been added. A distinction is made between native and exotic species in Appendix A.

Point 2: First two sentences of abstract do not fit together. Moreover, the first sentence is kind of eulogy of Chinese law. After reading this sentence one can make impression, that there is the most pro-environment country in the World, whereas there are numerous examples of ecologiacl disasters in China (but there are also spectacular successes). Plese revise.

Response 2: We agree with this suggestion and rewrite the summary section.

 “China is a big country in turtle cultivation breeding and has a long history of turtle breeding. In this study, the census and statistics of artificially domesticated aquatic turtles in 15 provinces of China were conducted, the results showed that 29 were native species of aquatic turtles in China, accounting for approximately 9% of the world's total, and a large number of exotic aquatic turtles are also domesticated in China. The present situation of artificial breeding and protection of aquatic turtles in major provinces of China is shown, and the existing problems are also put forward to provide suggestions for the protection and management of aquatic turtles.”

Point 3: Page 8. „Affairs of People's Republic China, 2020) . According to the growth data and the health” – there is space between parenthesis ‘)’ and a dot.

Response 3: Revised.

Reviewer 3 Report

This paper reports the current status of conservation of aquatic turtles in China, mainly in the viewpoint of political management. In my opinion, the context of this paper is too local and “analysis” commented in the title is insufficient. Especially, novelty was not found anywhere. Therefore, this paper seems not to have an appealing point to readers.

In abstract, the first sentence is unnecessary.

Author Response

Point 1: This paper reports the current status of conservation of aquatic turtles in China, mainly in the viewpoint of political management. In my opinion, the context of this paper is too local and “analysis” commented in the title is insufficient. Especially, novelty was not found anywhere. Therefore, this paper seems not to have an appealing point to readers.

Response 1: According the Comments of Reviewer 1 and Reviewer 2, they all felt that this paper had made an important contribution. Through investigation, this paper summarizes the current situation of breeding management in China, puts forward some problems and carries on corresponding discussions, which is of positive significance for the protection of Chinese turtles and turtles.

Point 2: In abstract, the first sentence is unnecessary.

Response 2: Yes, we rewrite the summary section.

Round 2

Reviewer 3 Report

This paper has been well revised. Therefore, I have no reason to reject this paper.